# The Impacts of Sun Exposure on Worker Physiology and Cognition: Multi-Country Evidence and Interventions

**DOI:** 10.3390/ijerph18147698

**Published:** 2021-07-20

**Authors:** Leonidas G. Ioannou, Lydia Tsoutsoubi, Konstantinos Mantzios, Giorgos Gkikas, Jacob F. Piil, Petros C. Dinas, Sean R. Notley, Glen P. Kenny, Lars Nybo, Andreas D. Flouris

**Affiliations:** 1FAME Laboratory, Department of Physical Education and Sport Science, University of Thessaly, 42100 Trikala, Greece; ioannouLG@gmail.com (L.G.I.); lydiatsoutsoubi@gmail.com (L.T.); konstantinosmantzios@gmail.com (K.M.); ggkikas@uth.gr (G.G.); petros.cd@gmail.com (P.C.D.); 2Department of Nutrition, Exercise and Sports, August Krogh Building, University of Copenhagen, 2100 Copenhagen, Denmark; jpp@nexs.ku.dk (J.F.P.); nybo@nexs.ku.dk (L.N.); 3Human and Environmental Physiology Research Unit, Faculty of Health Sciences, University of Ottawa, Ottawa, ON K1N 6N5, Canada; snotley@uottawa.ca (S.R.N.); gkenny@uottawa.ca (G.P.K.); 4Clinical Epidemiology Program, Ottawa Hospital Research Institute, Ottawa, ON K1H 8L6, Canada

**Keywords:** solar radiation, heat, occupational, labor, performance, core temperature, skin temperature, heart rate, skin blood flow, sweat rate

## Abstract

Background: A set of four case-control (*n* = 109), randomized-controlled (*n* = 7), cross-sectional (*n* = 78), and intervention (*n* = 47) studies was conducted across three countries to investigate the effects of sun exposure on worker physiology and cognition. Methods: Physiological, subjective, and cognitive performance data were collected from people working in ambient conditions characterized by the same thermal stress but different solar radiation levels. Results: People working under the sun were more likely to experience dizziness, weakness, and other symptoms of heat strain. These clinical impacts of sun exposure were not accompanied by changes in core body temperature but, instead, were linked with changes in skin temperature. Other physiological responses (heart rate, skin blood flow, and sweat rate) were also increased during sun exposure, while attention and vigilance were reduced by 45% and 67%, respectively, compared to exposure to a similar thermal stress without sunlight. Light-colored clothes reduced workers’ skin temperature by 12–13% compared to darker-colored clothes. Conclusions: Working under the sun worsens the physiological heat strain experienced and compromises cognitive function, even when the level of heat stress is thought to be the same as being in the shade. Wearing light-colored clothes can limit the physiological heat strain experienced by the body.

## 1. Introduction

The long-term health effects of sun exposure have been extensively studied, particularly in relation to skin cancer and cataract, but there is little evidence-based knowledge on the acute impacts of sun exposure. For instance, a laboratory study showed marked negative effects of solar radiation on human cognitive performance [1], but we do not know if the heat from the sun can affect the physiology and cognition of people working outdoors [2,3]. The World Health Organization and the International Labor Organization are developing joint methodologies for estimating the associated work-related burden of disease and injury. However, practical and economically feasible protection strategies for people working outdoors have not been investigated at the size and quality needed to draw robust conclusions and recommendations [4].

Agriculture and construction include the vast majority of employees exposed to the sun [2,3] due to the size of these industries, the requirement to work outdoors, and the lack of cost-effective shading solutions for these occupational settings. The associated societal and economic impacts are widespread. The agricultural sector alone, employing one-third of the world’s labor force [5], is projected to account for 60% of global working hours lost to heat stress in 10 years from now, whereas construction is expected to account for 19% of such loss [3]. These estimates translate to 63 million full-time jobs lost across the globe, with an associated monetary loss of US$ 1.9 trillion in purchasing power parity terms [3,6]. While these figures are astounding, they almost certainly underestimate the phenomena that will occur within the next 10 years for two reasons. First, these projections assume that the increase in global mean temperature at the end of the century will not exceed 1.5 °C (2.7 °F) of pre-industrial levels [3,7]. Unfortunately, it is now clear that global temperature is already 1.0 °C (1.8 °F) above pre-industrial levels, and it is likely to reach 1.5 °C between 2030 and 2052 if it continues to increase at the current rate [8]. This climate change is also expected to increase people’s exposure to the sun, particularly for those who work outdoors [9]. Second, they assume that work in agriculture and construction is carried out in the shade. However, recent data confirmed that these employees perform the vast majority of their work outdoors and are directly affected by sun exposure, leading to increased heat strain and impaired capacity for manual labor [10,11,12,13].

Based on these important knowledge gaps for addressing sun exposure and heat stress emergencies, the goals of this article are to provide health advisors and medical readers with evidence-based information on (i) the effects of sun exposure on worker physiology and cognition and (ii) a practical and economically feasible protection strategy for the most vulnerable individuals.

## 2. Materials and Methods

The goals of this paper are achieved by presenting relevant findings from a series of studies carried out in different industries and countries that include two large field trials, one randomized controlled trial in laboratory settings, and two field interventions. A set of four interconnected studies was conducted as follows:

### 2.1. Aims of the Studies

Study 1: The effects of solar radiation on the psychophysical stress experienced by workers who perform manual labor in construction and agriculture. The aim of this study was to investigate the effects of solar radiation on human psychophysical stress during actual work shifts in the heat.Study 2: The effects of solar radiation on physiological responses and cognitive function at rest and during physical work. The aim of this study was to perform a controlled, laboratory-based evaluation of the effects of solar radiation on human physiological responses and cognitive performance at rest, during physical work, and post-work recovery by comparing indoor (i.e., without solar radiation) and outdoor (i.e., typical mid-day solar radiation) environments characterized by the same thermal stress.Study 3: Identifying factors increasing the adverse effects of sun exposure experienced by agriculture and construction workers. The aim of this study was to investigate possible factors exacerbating the effects of sunlight-induced thermal strain by investigating workers’ behavioral habits during actual work shifts in occupational settings.Study 4: Interventions to mitigate the sunlight-induced heat strain experienced by people who work in agriculture and construction. The aim of this study was to test interventions to mitigate the sunlight-induced heat strain experienced by workers who work in agriculture and construction.

### 2.2. Experimental Protocol

Study 1: Detailed information on Study 1, including information about the physiological data we collected as well as supplemental tables and figures, is presented in Appendix A. This case-control study involved monitoring 109 experienced and heat acclimatized agriculture and construction workers during four to five consecutive full 11-h work shifts. Physiological, subjective, labor, and environmental data were collected throughout the study. Work hours were characterized by the same thermal stress, but different solar radiation levels were isolated to examine if solar radiation levels can independently modify the physiological heat strain experienced by workers.Study 2: Detailed information on Study 2, including information about the physiological data we collected as well as supplemental tables and figures, is reported in Appendix B. This single-blinded randomized controlled trial involved tracking seven participants during exposure to four different environmental conditions (two hot (30 °C wet-bulb globe temperature (WBGT) with and without solar radiation) and two temperate (20 °C WBGT with and without solar radiation) ambient conditions) allocated in random order. Physiological and subjective data were collected throughout the experiments. This study was conducted to confirm and complement the findings of Study 1 by delineating the physiological and cognitive impacts of sun exposure on people who perform manual work in environments characterized by the same thermal stress but different solar radiation levels.Study 3: Detailed information on Study 3 is overviewed in Appendix C. This cross-sectional study involved monitoring 78 agriculture workers from seven countries over a period of three months to examine the color of their clothing, a key factor mediating heat exchange, during actual work shifts performed outdoors.Study 4: Detailed information on Study 4 is outlined in Appendix D. This intervention study was conducted to investigate if changes in the color of workers’ clothing can modify the physiological heat strain experienced by people who work under the sun. The study involved monitoring 47 outdoor workers during two work shifts (“business as usual” and “white clothing” scenarios) characterized by the same thermal stress and solar radiation levels. Physiological, labor, and environmental data were collected to investigate if white clothes can reduce the physiological heat strain experienced by people who work under the sun.

## 3. Results

Sun exposure increases the physiological heat strain experienced by workers and compromises their cognitive function, even when the level of heat stress is thought to be the same as being in the shade. An overview of our findings is presented in Figure 1, while detailed information is provided under the subheading dedicated to each study.

### 3.1. Results of Study 1: Effects of Solar Radiation on the Psychophysical Stress Experienced by Workers who Perform Manual Labor in Construction and Agriculture

A group of 109 workers (Table A1) participated in the study. During the study, 396 h (collected from 98 workers) were identified as having equal thermal stress (30 °C WBGT) but different solar radiation levels (ranging from 0 W/m^2^ to 1043 W/m^2^). Of the 396 h, 108 (27.3%) were successive hours. Solar radiation levels were positively associated with mean skin temperature (T_sk_) (r = 0.419, *p* < 0.001; Figure 2). This association was characterized by an increase of ~0.2 °C for every 100 W/m^2^ increase in solar radiation levels (Figure 2). On the other hand, a negligible negative association was found between solar radiation levels and core body temperature (T_core_) as estimated by gastrointestinal temperature (r = −0.141, *p* = 0.035; Figure A1). Additionally, no associations were identified between solar radiation levels and heart rate (Figure A2) or metabolic rate (Figure A3). Perceived thermal radiation was significantly related to specific items of the Heat Strain Score Index [16] evaluating subjective psychophysical parameters (Table A2). Moreover, significant differences in T_sk_ (Figure 2) and labor intensity (Figure A3) were identified between indoor (0 to 160 W/m^2^), mixed (161–320 W/m^2^), and outdoor (>320 W/m^2^) environments (F_(2, 360)_ = 57.791, *p* < 0.001).

Although we tested the same workers working in environments characterized by the same thermal stress (30 °C WBGT), having a Heat Strain Score Index greater than 18 (indicating dangerously high risk of experiencing heat strain) was 3.61 times more likely when working outdoors as compared to indoors (Figure 1). Even more so, having a T_sk_ above 36 °C (indicating dangerously high risk of experiencing heat strain [14,15]) was 10.16 times more likely when working outdoors as compared to indoors (Figure 1). Similarly, the risk for experiencing dizziness, weakness, and other heat strain symptoms (i.e., mild headache, muscle pain, the appearance of red acne, and reduced mental concentration) are 4.44, 3.17, and 2.40 times higher, respectively, when working outdoors compared to indoors (Table A3).

### 3.2. Results of Study 2: Effects of Solar Radiation on Physiological Responses and Cognitive Function at Rest and during Physical Work

The anthropometric characteristics of the seven volunteers that participated in the study were as follows: age: 22.7 ± 3.2 years; body stature: 177.6 ± 6.1 cm; body mass: 74.3 ± 8.9 kg; body fat: 20.1 ± 6.7%; and lean mass: 56.7 ± 5.1%. Although participants were exposed to environments characterized by equal thermal stress, we identified that exposure to solar radiation had an incremental effect on the mean skin temperature (Figure 3 and Figure 4) of the participants impairing their cognitive performance (Table A5). Furthermore, we found that cognitive performance was positively (i.e., participants made more mistakes when their T_sk_ and T_core_ were increased) related to T_sk_ and T_core_ (Table A6).

### 3.3. Results of Study 3: Identifying Factors Increasing the Adverse Effects of Sun Exposure Experienced by Agriculture and Construction Workers

A group of 78 agriculture workers (age: 42.4 ± 13.0 years; height: 166.1 ± 11.5 cm; and weight: 70.5 ± 19.6 kg) was monitored for a total of 112 work shifts over a period of three months. We identified that more than two-thirds (68.8%) of the monitored workers wore dark-colored clothes during work under the sun. This was recognized as an important factor increasing the thermal strain experienced by workers. It is important to note that this is against prevailing recommendations and may reflect the lack of knowledge of workers and employers regarding occupational health and safety.

### 3.4. Results of Study 4: Interventions to Mitigate the Sunlight-Induced Heat Strain Experienced by Workers Who Work in Agriculture and Construction

A group of six agriculture (two females) and 41 construction (all males) workers participated in the study (Table A7). The two scenarios (“business as usual” (BAU) and “white clothing” (CLO)) were characterized by similar thermal stress (agriculture: ~23 ± 4 °C WBGT; and construction: ~28 ± 4 °C WBGT) and solar radiation levels (agriculture: ~920 ± 300 W/m^2^; and construction: ~200 ± 150 W/m^2^). Moreover, task analysis identified that workers performed similar manual work (agriculture (BAU: 194 ± 54 W/m^2^ vs. CLO: 189 ± 53 W/m^2^) and construction (BAU: 96 ± 17 W/m^2^ vs. CLO: 86 ± 17 W/m^2^)) during both scenarios. We identified no significant differences in the T_core_ of workers between the tested scenarios (agriculture (BAU: 37.3 ± 0.3 °C vs. CLO: 37.2 ± 0.3 °C) and construction: (BAU: 37.3 ± 0.2 °C vs. CLO: 37.4 ± 0.2 °C)). Importantly, although workers were exposed to environments characterized by the same thermal stress and doing the same labor, workers donning white uniforms experienced a reduced level of physiological heat strain (i.e., small reductions in mean skin temperature and some minor improvement in thermal sensation; Table A8). Furthermore, the average change in T_sk_ (from resting conditions) was 13% and 12% lower during the white clothing scenario compared to the business-as-usual scenario in agriculture and construction, respectively.

## 4. Discussion

To investigate the physiological and health impacts of sun exposure on workers performing jobs/duties outdoors, a series of four separate but interconnected studies were conducted across different industrial sectors and countries. A large-scale occupational field trial was conducted in Qatar involving 109 construction and agricultural workers. From the collected data, we compared individuals working in the shade versus those working under the sun. We also isolated 396 work hours characterized by the same high level of heat stress (30 °C WBGT; typical for a Northern hemisphere heatwave) but very different levels of sun exposure: 33% were in the shade, and 67% were under the sun. We found that people working under the sun were four times more likely to experience dizziness (i.e., vertigo, presyncope, disequilibrium, or other non-specific feelings), three times more likely to report weakness, and twice more likely to suffer other symptoms of heat strain (i.e., mild headache, muscle pain, the appearance of red acne, and reduced mental concentration), compared to performing the same work in the shade under the same level of heat stress. These clinical impacts of sun exposure were not accompanied by changes in core body temperature but, instead, were linked with changes in skin temperature, which was 10 times more likely to be at levels indicative of heat strain (>36 °C) when working under the sun.

Skin temperature is an important parameter linked with both physiological and psychophysical stress [14,15,17]. An increasing number of reports over the last decade have highlighted the importance of high skin temperature as an early indicator of hyperthermia and heat injury, as well as for regulating the intensity of work and exercise [13,17,18,19,20,21,22]. Recent data from the European Commission project HEAT-SHIELD [23] show that higher skin temperature is linked with reduced capacity to perform manual labor, leading to significant economic losses [13,24]. Overall, the findings from the first study show that working under the sun increases skin temperature and the risk for experiencing clinical symptoms of heat strain, albeit without markedly altering physiological heat strain as defined by changes in core temperature and heart rate, even in cases where the level of environmental heat stress is considered to be the same as working in the shade. This is probably related to the well-described effect of self-pacing that is known to act proactively to avoid an increase in workers’ core body temperature [13,25]. However, self-pacing may not be appropriate when jobs or tasks are time-sensitive, involve productivity incentives, and/or involve workers who are not well trained in their job [25,26].

To delve deeper into the physiological and cognitive impacts of sun exposure, as well as to better position the findings of our field experiments, we conducted in Greece a randomized controlled trial wherein seven healthy individuals were monitored during rest, moderate-intensity physical work, and post-work recovery inside a climate-controlled chamber. We compared values when participants were under the sun versus in the shade in temperate (20 °C WBGT) and hot (30 °C WBGT) ambient conditions. This study confirmed that sun exposure could elevate skin temperature without affecting core body temperature. Other physiological responses (heart rate, skin blood flow, and sweat rate) were also increased during sun exposure but to a smaller degree. More importantly, sun exposure reduced cognitive performance in both temperate and hot conditions, leading to a 45% reduction in divided attention (e.g., auditory and visual stimuli in parallel) and a 67% reduction in vigilance tasks (see Appendix B for measurement details). Literature suggests that environmental heat stress increases the risk of occupational injuries by promoting fatigue, reduced psychomotor performance, loss of concentration, and reduced alertness [27]. In total, the findings from the second study confirmed that sun exposure generates symptoms of heat strain, such as dizziness and weakness, and it also undermines cognitive function in both temperate and hot conditions, even when the level of heat stress is thought to be the same as being in the shade.

A practical and cost-effective strategy to limit the impact of sun exposure is to change the color of the clothing worn. Wearing white or light-colored clothes increases the reflection of heat from the sun [28,29,30,31,32] and can limit the heat strain (e.g., skin temperature, heart rate, and sweat rate) that the human body experiences during physical work in a hot environment [29,31]. Therefore, several heat stress guidelines recommend wearing light-colored clothes during work or exercise under the sun [30,33,34,35,36].

To assess attitudes of outdoor workers in relation to the choice of color of their clothing, we monitored 78 workers originating from seven countries (Bangladesh, Cyprus, Egypt, India, Philippines, Romania, and Vietnam) during 112 full work shifts performed outdoors in Cyprus during summer and autumn. Overall, the findings from the third study showed that 68.8% of the studied outdoor workers wore dark-colored clothes. This is against prevailing recommendations and reflects the lack of education and training on heat-related aspects of occupational health and safety for both workers and employers. Dark-colored clothes have high absorbance of radiative heat, and it is likely that workers would benefit from changing to light-colored work uniforms to minimize the adverse effects of sun exposure.

In an occupational field intervention study performed in Cyprus, we tested whether white clothes can be used as a practical and economically feasible strategy to limit the impacts of sun exposure in agricultural workers. We monitored six workers during two full work shifts characterized by temperate conditions. In the “business as usual” scenario, workers wore their preferred clothes. As expected, based on the results of the above-described observational study, these clothes were mostly black or dark-colored t-shirts and pants. In the “white clothes” scenario, workers were provided with white hats and t-shirts (all 100% cotton; total cost: US$ 8.80) and were instructed to wear light-colored pants of their own. The findings from the fourth study demonstrated that the change to white/light-colored clothing minimized the increase in mean skin temperature during work by 13% (corresponding to a reduction of 0.4 °C), despite that the participants worked at the same level of effort in the same outdoor conditions.

In a second occupational field intervention performed as part of our above-mentioned occupational field trial in Qatar [11], we tested the efficacy of wearing white coveralls for limiting the impacts of sun exposure in construction workers. We monitored 41 construction workers during two full work shifts characterized by moderate heat stress. In the “business as usual” scenario, workers wore their typical work coveralls (dark blue color in most cases) made of polycotton. In the “white coveralls” scenario, workers were provided with white coveralls (total cost: US$ 6.00) made of cotton or polycotton. This change to white coveralls minimized the increase in mean skin temperature during work by 12% (corresponding to a reduction of 0.2 °C), despite that the participants worked at the same level of effort in the same outdoor conditions. Taken together, these results support existing recommendations for wearing light-colored clothes during work or exercise under the sun [30,33,34,35,36] and further demonstrate their practicality and cost-effectiveness in occupational settings. While white work coveralls can be effective in reducing the physiological strain during work under the sun, the adoption of other heat mitigation strategies, such as hydration protocols, work–rest cycles for jobs that do not allow for self-pacing, ventilated garments, and mechanization of heavy work, can further reduce the physiological strain experienced by people working manually outdoors [25].

## 5. Conclusions

Medical staff are often asked whether occupational injuries are caused by exposure to the sun per se or by other parameters of heat stress such as high temperature or humidity. The field studies and laboratory-based clinical trials that we conducted in different parts of the world under both temperate and hot conditions show that working under the sun worsens the physiological heat strain experienced and compromises cognitive function, even when the level of heat stress is thought to be the same as being in the shade. To limit these detrimental impacts of sun exposure in cases where no other cost-effective shading solutions are available, medical staff, as well as health and safety professionals, should advise outdoor workers to wear white or light-colored clothes and hats/helmets. These pale colors increase the reflection of heat from the sun and can limit the heat strain experienced by the body. This multi-country series of field, clinical, and intervention studies raises another important issue; guidelines and policies should consider sun exposure as an important modifier of occupational and public health, not as one mere physical element to be included in the calculation of heat stress. Occupational and public health guidelines should be adapted based on exposure to solar radiation, fueled by a much-needed estimation of the associated burden of disease and injury.

## Figures and Tables

**Figure 1 ijerph-18-07698-f001:**
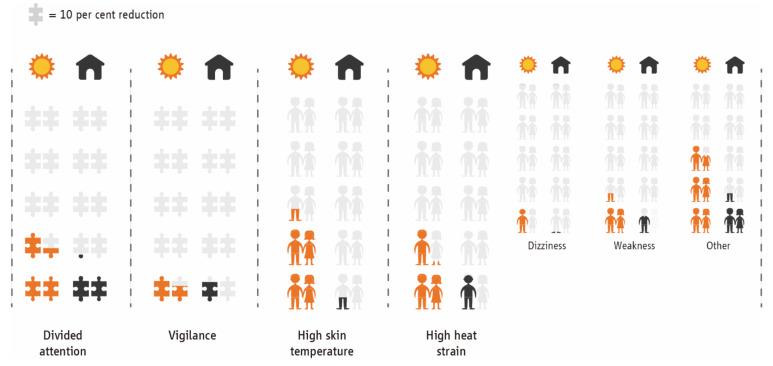
Cognitive and health impacts while working under the sun (left column) and in the shade (right column) in a hot environment (30 °C WBGT). Each full-colored puzzle piece indicates a 10% reduction in cognitive performance (divided attention and vigilance). Each full-colored body figure indicates one-in-ten workers experiencing mean skin temperature higher than 36 °C (the threshold for progressive symptoms of heat strain [14,15]) or danger-level heat strain, including dizziness, weakness, or other symptoms.

**Figure 2 ijerph-18-07698-f002:**
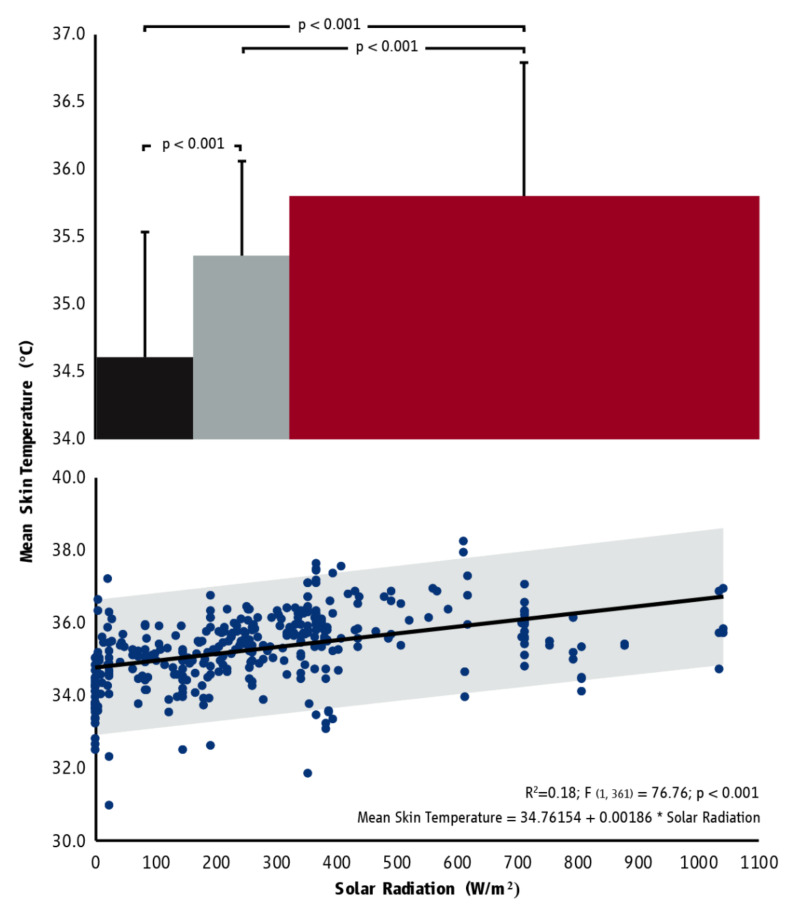
The average (±SD) mean skin temperature in indoor (black), mixed (grey), and outdoor (red) environments characterized by the same thermal stress (30 °C WBGT) but different solar radiation levels (top graph) as well as the association between hourly mean skin temperature and solar radiation (bottom graph). The bar width in the top graph indicates the range of solar radiation of each category corresponding to the horizontal axis, while horizontal brackets indicate statistically significant differences. Shading in the bottom graph corresponds to the 95% prediction interval.

**Figure 3 ijerph-18-07698-f003:**
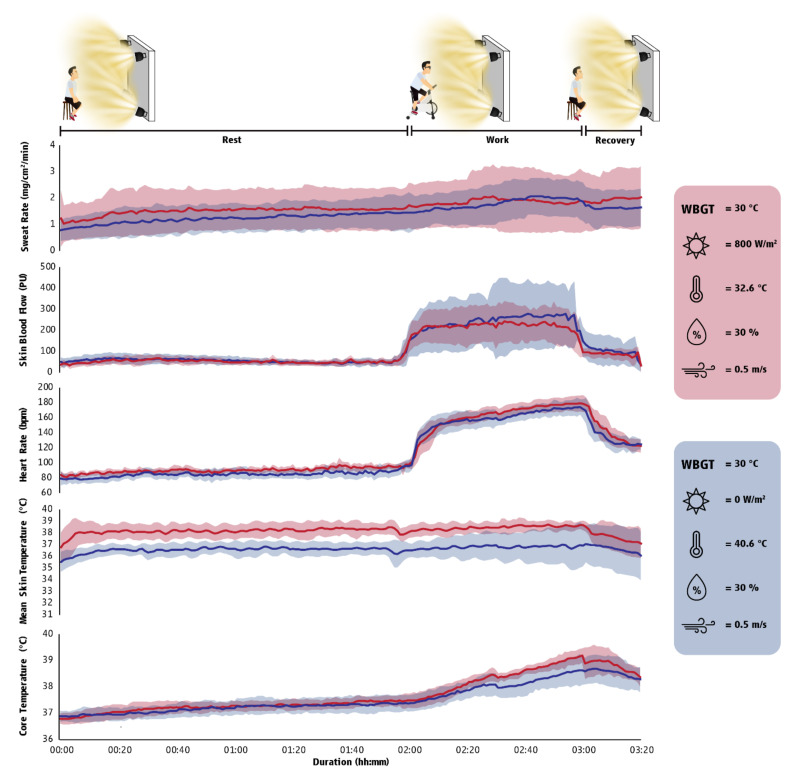
Physiological responses (mean ± SD) during exposure in a hot environment (30 °C WBGT). The first two hours (00:00 to 02:00) illustrate the fluctuation of physiological responses in resting conditions, the third hour (02:00–03:00) illustrate physiological responses during exercise/work, while the final 20 min (03:00–03:20) show the responses during recovery time. Red indicates a hot outdoor environment, while blue indicates a hot indoor environment. Sweat rate corresponds to the average sweat rate from the forehead, arm (bicep), and thigh (quadricep) as measured using the ventilated capsule technique, expressed in milligrams per centimeter square per minute. Skin blood flow as measured by laser-Doppler flowmetry corresponds to the average skin blood flow from the forearm (brachioradialis) and leg (gastrocnemius), expressed in perfusion units. Heart rate is expressed in beats per minute. Mean skin temperature estimated from arm, chest, thigh, and leg skin temperatures, expressed in degrees Celsius. Core temperature corresponds to gastrointestinal temperature, expressed in degrees Celsius. Effect sizes for all comparisons between outdoor and indoor environments can be found in Table A5.

**Figure 4 ijerph-18-07698-f004:**
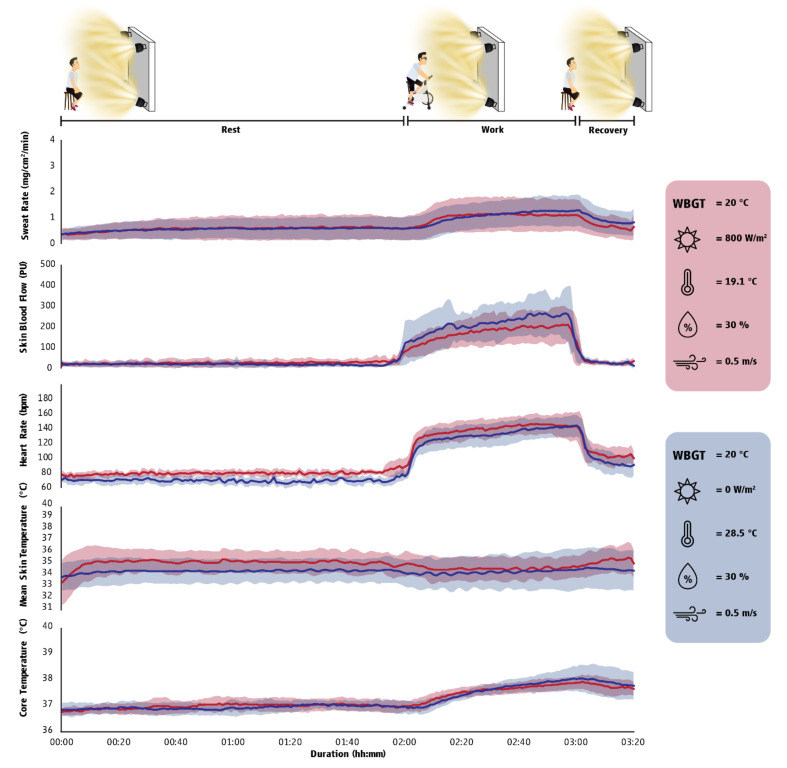
Physiological responses (mean ± SD) during exposure in a temperate environment (20 °C WBGT). The first two hours (00:00 to 02:00) illustrate the fluctuation of physiological responses in resting conditions, the third hour (02:00–03:00) illustrate physiological responses during exercise/work, while the final 20 min (03:00–03:20) show the responses during recovery time. Red indicates a thermoneutral outdoor environment, while blue indicates a thermoneutral indoor environment. Sweat rate corresponds to the average sweat rate from the forehead, arm (bicep), and thigh (quadricep) as measured using the ventilated capsule technique, expressed in milligrams per centimeter square per minute. Skin blood flow as measured by laser-Doppler flowmetry corresponds to the average skin blood flow from the forearm (brachioradialis) and leg (gastrocnemius), expressed in perfusion units. Heart rate is expressed in beats per minute. Mean skin temperature estimated from the arm, chest, thigh, and leg skin temperatures, expressed in degrees Celsius. Core temperature corresponds to gastrointestinal temperature, expressed in degrees Celsius. Effect sizes for all comparisons between outdoor and indoor environments can be found in Table A5.

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
