# Peer review of "The Impacts of Sun Exposure on Worker Physiology and Cognition: Multi-Country Evidence and Interventions"

_ijerph, 2021, doi:10.3390/ijerph18147698_

Round 1

Reviewer 1 Report

I do not have any further comments.
Manuscript may be accepted.

Author Response

Thank you for your constructive comments which helped us to improve the overall quality of our paper.

Reviewer 2 Report

The authors provide findings from four studies that, taken together, support the conclusion that work in the sun is associated with higher skin temperatures than comparable work in the shade. Comparable includes similar metabolic heat and the same environmental heat level determined by equal WBGT. Effects in addition to increased skin temperature include worker reports of increase in feeling some dizziness, weakness, and other indices of heat strain. The studies were conducted in Qatar, Greece, and Cyprus. One of the field studies documented the benefits of wearing light-colored clothing when working in the sun in terms of limiting increased skin temperature compared to wearing dark-colored clothing. In other studies, increased skin temperature has been associated with signs of heat disorders.

The overall organization of the article is to present the larger picture in the main paper with four appendices describing the specifics of the four studies. In my opinion, the article describes an impressive set of studies that used different experimental approaches for supporting the two stated goals of providing evidence-based information on (i) the effects of sun exposure on worker physiology and cognition, and (ii) a practical and economically feasible protection strategy for the most vulnerable individuals.

I suggest a small change in the Abstract. The use of second person (we) is alright in the body of a paper, but I think it is inappropriate in an Abstract. In line 17 I suggest replacing We with “The authors…”

Author Response

Comment 1: The authors provide findings from four studies that, taken together, support the conclusion that work in the sun is associated with higher skin temperatures than comparable work in the shade. Comparable includes similar metabolic heat and the same environmental heat level determined by equal WBGT. Effects in addition to increased skin temperature include worker reports of increase in feeling some dizziness, weakness, and other indices of heat strain. The studies were conducted in Qatar, Greece, and Cyprus. One of the field studies documented the benefits of wearing light-colored clothing when working in the sun in terms of limiting increased skin temperature compared to wearing dark-colored clothing. In other studies, increased skin temperature has been associated with signs of heat disorders.

The overall organization of the article is to present the larger picture in the main paper with four appendices describing the specifics of the four studies. In my opinion, the article describes an impressive set of studies that used different experimental approaches for supporting the two stated goals of providing evidence-based information on (i) the effects of sun exposure on worker physiology and cognition, and (ii) a practical and economically feasible protection strategy for the most vulnerable individuals.

I suggest a small change in the Abstract. The use of second person (we) is alright in the body of a paper, but I think it is inappropriate in an Abstract. In line 17 I suggest replacing We with “The authors…”

Response 1: Thank you for your supportive comments. We agree with the reviewer and our abstract has been now changed as follows:

L 17-19: “Physiological, subjective, and cognitive performance data were collected from people working in ambient conditions characterized by the same thermal stress but different solar radiation levels.”

Reviewer 3 Report

General

The structure and concept of this manuscript is different compared to what this reviewer is used to.

For example, Conclusions start with

"About one fatal case of occupational heat stress is expected to occur every 14 to 24 minutes during the next decade [32]. Heat stress also causes many occupational injuries by promoting fatigue, reduced psychomotor performance, loss of concentration, and attenuated alertness [22]."

This is not a conclusion based on the present manuscript!?

Overall, the authors do not seem to be aware of the scientific literature and recommendations from Occupational Safety and Health bodies regarding work in high temperature!? They need at least to refer to the present recommendations and comment if they are not supported by enough scientific evidence. Then this can motivate more studies including this one to develop them further. The use of light cotton clothes has for example been integrated in the recommendations since many years. 

The paper has similarities with a potential section in a book. If the authors want to cover the subjects at large. Then I think they also need to comment on dehydration aspects. See for example Glaser J, Hansson E, Weiss I, et al. Occup Environ Med 2020;77:527–534. These authors have focused on Kidney effects among sugar cane workers.

Aims

Study 1: Effects of solar radiation on the psychophysical stress experienced by workers who perform manual labor in construction and agriculture. The aim of this study was to investigate the effects of solar radiation on human psychophysical stress during actual work shifts in the heat.

They found an increase skin temperature and according to me also a tendency of decreased core temperature <0.05! It seems reasonable to me that sun radiation heats the surface but why do core temperature decrease? Do the study subjects compensate by drinking or anything else?

Study 2: Effects of solar radiation on physiological responses and cognitive function at rest and during physical work. I agree with the interpretation of the difference in effect between exposure at sun-light and comparison. For preventive purposes I think it would be of value to comment on the almost monotonous increase in heart rate and core temperature. Explain why do you focus on skin temperature. It might be related to sympoms of body warning but do you have support for that this is the best measure to intervene on?

Comment that there also is a health side above from the effects you choose to focus on. You do not comment anything regarding how much and at what intensity the workers are recommended to work. See enclosed file.

Study 3: Identifying factors increasing the adverse effects of sun exposure experienced by agriculture and construction workers.

"We identified that more than two thirds (68.8%) of the monitored workers wore dark-colored clothes during work under the sun. " 

This is against prevailing recommendations can you comment on why dark clothes are dominating?

Study 4: Interventions to mitigate the sunlight-induced heat strain experienced by people who work in agriculture and construction. The aim of this study was to test interventions to mitigate the sunlight-induced heat strain experienced by workers who work in agriculture and construction. 

The findings from the fourth study demonstrated that the change to white /  light colored clothing minimized the increase in mean skin temperature during work by 13% (corresponding to a reduction of 0.4°C).  This is not a major effect. I miss information on work intensity this might be important to understand if there is potential to find and effect. It might be more important at work intensity.

Round 2

Reviewer 3 Report

The manscript has improved significantly. Still i do not follow the ölogis on lines 269-277 new version.

To delve deeper into the physiological and cognitive impacts of sun exposure, as well  as to better position the findings of our field experiments, we conducted in Greece a randomized controlled trial wherein seven healthy individuals were monitored during rest, moderate intensity physical work, and post-work recovery inside a climate-controlled chamber. We compared values when participants were under the sun versus in-shade in temperate (20°C WBGT) and hot (30°C WBGT) ambient conditions. This study confirmed  that sun exposure can elevate skin temperature without affecting core body temperature. Other physiological responses (heart rate, skin blood flow, and sweat rate) were also in creased during sun exposure, yet to a smaller degree. 

Lokking at figure 3 and to some extent also in figure 4 Core temperature do increase during work. Do _I misunderstand something here? 

Author Response

This manuscript is a resubmission of an earlier submission. The following is a list of the peer review reports and author responses from that submission.

Round 1

Reviewer 1 Report

This article concerns an evaluation about sun exposure to workers health. The text appears to be unbalanced in its underlying structure. In my opinion, the paper needs a thorough revision and re-editing. Some observations (not exhaustive):

  • I suggest to revise the title by introducing which area of ​​work has been studied;
  • The abstract does not clarify the 4 research works carried out individually. The 241 enrolled workers of different studies are brought together. The abstract should be more impactful and there is a lack of statistical information;
  • There are several standardized heat assessment models that are not described in the introduction and have not been considered for various comparisons;
  • Have the 4 works presented already been published?
  • The 4 research works seem to have been constructed for slightly different purposes: how is it possible to aggregate the conclusions?
  • The discussion needs a thorough review in order to clarify how the data was aggregated to have a structured overview of the results collected;
  • Probably the collected data in the 4 experimental studies can be revised, expanded and divided into different research articles.

Reviewer 2 Report

The authors present the results of two studies evaluating the physiological, psychological effects (psychophysical stress) of solar radiation. The first study was cross-sectional, a second one was a single-blind randomized controlled trial

Two others studies assessed the protection provided to workers by wearing white cloths. First of them has monitored,  over a period of three months, workers from 7 countries clothing in association with physiological response to solar radiation. The second one investigated the effectiveness of white clothing protection in the agriculture and construction industry.

The authors present an enormous dose of information mainly providing details of carried out measurements. I would strongly suggest eliminating a major part of this data. Only key information regarding the number of subjects and conditions the observations were done should be reported. One of the options would be to put the methodological details to the Annex placed on the journal website.

I would eliminate also graphs 4,5 and 6 as they illustrate “negative” findings.

Reviewer 3 Report

The authors provide an analysis of four studies to assess the impact of sun exposure on worker physiology and cognition.

Two of the studies are extremely small 7 and 6 participants respectively and the only analysis possible is a descriptive one. The effort of the authors to further analysis the data is exaggerate and technically questionable.

Overall the manuscript is informative.

Some comments

Lines 24-25 …reduced by 45% to 67%..’ compared to what? What is the baseline?

Lines 25-26 ‘ …12 to 13%..’ compared to what? Baseline?

Lines 142-144. ‘…according to Google scholar metrics..’ This is an interesting but non-scientific argument. Because a metric has many hits does not mean that it is right. Please provide scientific arguments for your choice.

Line 198. Seven participants is an extremely small sample. Therefore I would only present descriptive statistics.

Lines 258-266. The authors identify an effect size of 1.27 and take it as guidance for the sample size of seven!!! Participants. Even if the calculation is right common sense says that seven participants can never be used to provided inferential statistics. Even a Person correlation coefficient is meaningless in this case. Simple descriptive stats should be enough to describe this studies results. Please adapt methods and results accordingly.

Line 381. This study had 6 participants. Therefore the very small sample does not allow for more that simple descriptive statistics again. Please adapt methods and results accordingly.

Table 3. The correlations performed in Table 3 lead to the question of Bonferroni adaptation. Did you consider it?

Figure 3-6.What is the slope of the regression lines in these calculations? The authors should describe the results on the basis of the slope and not the less informative correlation coefficient.

Table 5. Again there is an issue with two many test. The p-values should be adjusted.

The reference list is missing